DOI: 10.1038/s41467-018-06301-2　　**OPEN**

# Apoε4 disrupts neurovascular regulation and undermines white matter integrity and cognitive function

Kenzo Koizumi[1], Yorito Hattori[1], Sung Ji Ahn[1], Izaskun Buendia[1], Antonio Ciacciarelli [1], Ken Uekawa[1], Gang Wang[1], Abigail Hiller[1], Lingzhi Zhao[1], Henning U. Voss [2], Steven M. Paul[3], Chris Schaffer [1,4], Laibaik Park [1] & Constantino Iadecola[1]

The ApoE4 allele is associated with increased risk of small vessel disease, which is a cause of vascular cognitive impairment. Here, we report that mice with targeted replacement (TR) of the ApoE gene with human ApoE4 have reduced neocortical cerebral blood flow compared to ApoE3-TR mice, an effect due to reduced vascular density rather than slowing of micro-vascular red blood cell flow. Furthermore, homeostatic mechanisms matching local delivery of blood flow to brain activity are impaired in ApoE4-TR mice. In a model of cerebral hypoperfusion, these cerebrovascular alterations exacerbate damage to the white matter of the corpus callosum and worsen cognitive dysfunction. Using 3-photon microscopy we found that the increased white matter damage is linked to an enhanced reduction of microvascular flow resulting in local hypoxia. Such alterations may be responsible for the increased susceptibility to hypoxic-ischemic lesions in the subcortical white matter of individuals carrying the ApoE4 allele.

[1] Feil Family Brain and Mind Research Institute, Weill Cornell Medicine, New York 10065 NY, USA. [2] Department of Radiology, Weill Cornell Medicine, New York 10065 NY, USA. [3] Department of Neurology and Psychiatry, Washington University in St. Louis, St. Louis 63110 MO, USA. [4] Meinig School of Biomedical Engineering, Cornell University, Ithaca 14853 NY, USA. These authors contributed equally: Kenzo Koizumi, Yorito Hattori. Correspondence and requests for materials should be addressed to L.P. (email: lap2003@med.cornell.edu) or to C.I. (email: coi2001@med.cornell.edu)

Microvascular alterations leading to hypoxic-ischemic white matter (WM) damage (small vessels disease, SVD) are the major cause of cognitive impairment on vascular basis[1], and have also been implicated in cognitive dysfunction and amyloid-β accumulation in Alzheimer's disease (AD)[2–4]. The pathogenic mechanisms by which SVD leads to WM damage remain to be established. Supplied by terminal arterioles in overlapping arterial territories, the subcortical WM is thought to be particularly vulnerable to vascular dysfunction and reduced cerebral perfusion[5]. However, little is known about the microcirculation of the deep cortical WM and the impact that cerebral hypoperfusion exerts on WM microvascular flow.

Although vascular risk factors, hypertension in particular, are well known causes of small vessel diseases and WM damage[6], genetic factors also play a role[7]. In particular, the ApoE4 allele, the leading genetic risk factor for sporadic AD, has also emerged as a risk factor for SVD and cognitive impairment on vascular basis[8–10]. Thus, individuals homozygous for ApoE4 have 3–4 fold increased risk of WM lesions, independently of other risk factors, such as age and hypertension, or AD diagnosis[11–13]. An association with microbleeds has also been reported[14], but it is unclear if it reflects cerebral amyloid angiopathy, for which ApoE4 is also a risk factor[15], possibly by impairing Aβ vascular clearance[16].

The mechanisms by which the ApoE4 may promote WM damage remain to be explored. Healthy individuals carrying the ApoE4 allele have reduced cerebral blood flow (CBF)[17], whereas mice with targeted replacement of ApoE with human ApoE4 have altered permeability of the blood brain barrier (BBB), an effect attributed to age-dependent pericyte loss[18]. However, whether ApoE4 alters microvascular function and promotes WM ischemic lesions remains to be established.

In this study, we used ApoE4-targeted replacement mice (ApoE4-TR) to gain insight into the increased susceptibility to WM damage conferred by the ApoE4 allele. We found major alterations in microvascular regulation in ApoE4-TR mice, associated with an increased susceptibility to WM damage and cognitive impairment in cerebral hypoperfusion produced by bilateral carotid artery stenosis (BCAS). Microvascular imaging with 3-photon microscopy revealed a remarkable susceptibility of WM microvascular flow to hypoperfusion leading to WM hypoxia and damage. The findings provide insight into the microvascular events underlying hypoxic-ischemic damage of the deep cortical WM and unveil a potential mechanism for the increased WM lesion burden in ApoE4 carriers.

## Results

**Resting CBF is reduced in ApoE4-TR mice.** First, we examined the impact of ApoE4 on baseline cerebral perfusion. To this end, we used ASL-MRI to measure CBF quantitatively (ml/100 g/min) in wild-type (WT), ApoE3-TR and ApoE4-TR mice. No difference in mean arterial pressure was found between WT [83 (mean) ± 3 (SEM) mmHg], ApoE3-TR (84 ± 3 mmHg) and ApoE4-TR mice (81 ± 2 mmHg). In ApoE3 mice somatosensory cortex CBF was not different from that of WT mice (Fig. 1a), but was reduced by 19 ± 4% in ApoE4-TR vs. WT mice and by 13 ± 5% vs. ApoE3-TR mice (Fig. 1a; $p < 0.05$). CBF was also reduced in the caudate nucleus in ApoE4-TR vs. WT mice ($-14 ± 4\%$ Supplementary Fig. 1A; $p < 0.05$). To provide insight into the microvascular bases of the reduction in resting CBF, we used in vivo 2-photon microscopy to assess microvascular diameter and red blood cell (RBC) speed in the neocortex. We focused on the somatosensory cortex, the same area where CBF was measured by ASL-MRI, under the same conditions of anesthesia and physiological parameters. Microvascular networks (arterioles,

capillaries, venules) were identified by morphological criteria and direction of flow and reconstructed[19]. We found that vascular diameter and RBC velocity in these microvessels did not differ between ApoE3-TR and ApoE4-TR mice (Fig. 1b).

To determine if the reduction in neocortical CBF was the result of reduced vascular density, we examined microvascular density using the endothelial markers CD31 and Glut-1. We found a reduction in the number and area of CD31-positive microvessels in the neocortex and corpus callosum (CC) of ApoE4-TR mice compared to WT and ApoE3-TR mice (Fig. 1c; $p < 0.05$). A reduction in vascular density in neocortex was also observed with Glut-1 as a marker (Supplementary Fig. 1B), an effect independent of neocortical atrophy assessed by measurement of cortical thickness with MRI (Supplementary Fig. 1C). As previously reported[18], pericyte coverage of cortical microvessels, assessed by CD13 immunocytochemistry, was reduced in ApoE4-TR compared to WT and ApoE3-TR mice (Supplementary Fig. 2A–D). Therefore, vascular density and resting cerebral perfusion are reduced in ApoE4-TR mice, which, in the absence of blood pressure changes, suggests an increase in cerebrovascular resistance compared to ApoE3-TR mice.

**Radical-dependent neurovascular dysfunction in ApoE4-TR mice.** Owing to limited energy stores, the brain relies on a continuous delivery of oxygen and glucose through blood flow, and, consequently, critical regulatory mechanisms assure that the brain's blood supply is well matched to the local metabolic needs of the tissue[20]. Alterations in CBF regulation cause brain dysfunction, often resulting in cognitive impairment[5]. Therefore, we investigated if key mechanisms regulating CBF, such as the coupling between neural activity and CBF, and the endothelial regulation of microvascular flow, are altered by ApoE4. ApoE3-TR, ApoE4-TR, and WT mice were equipped with a cranial window overlying the whisker barrel cortex and CBF responses to neural activity (whisker stimulation) and to neocortical superfusion with the endothelium-dependent vasoactive agents acetylcholine (ACh) and bradykinin (BK) were tested using laser-Doppler flowmetry (LDF). CBF responses to superfusion of the smooth muscle relaxants adenosine and the nitric oxide (NO) donor S-nitroso-N-acetylpenicillamine (SNAP), as well as hypercapnia, a potent vasodilatatory stimulus, were also examined. We found that the increase in somatosensory cortex CBF evoked by whisker stimulation, as well as those produced by neocortical application of ACh or BK was markedly reduced in ApoE4-TR mice (Fig. 2a–c; Supplementary Fig. 3A). In contrast, responses to adenosine, SNAP, and hypercapnia were not affected (Fig. 2d; Supplementary Fig. 3B–C), attesting to the selectivity of the effect of ApoE4 on endothelium-dependent responses and neurovascular coupling. The attenuation in neurovascular coupling is unlikely to result from an impairment of glutamate receptor function and reduced $Ca^{2+}$ influx, critical drivers of the CBF increase evoked by neural activity in the whisker barrel cortex[21]. This is because the increase in intracellular $Ca^{2+}$ induced by the glutamatergic agonist N-methyl-D-aspartate (NMDA) in isolated neocortical neurons was comparable between WT, ApoE3-TR and ApoE4-TR mice (Supplementary Fig. 4). Similarly, the presynaptic marker synaptophysin was not reduced in the neocortex of ApoE4-TR mice (Supplementary Fig. 5). Since oxidative stress has been associated with SVD in humans[22] and ApoE4 increases production of reactive oxygen species (ROS)[23], we examined the role of oxidative stress in the neurovascular dysfunction of ApoE4-TR mice. Neocortical superfusion with the ROS scavenger MnTBAP for 30–45 min did not ameliorate resting CBF, but completely rescued neurovascular coupling and endothelium-dependent responses to ACh

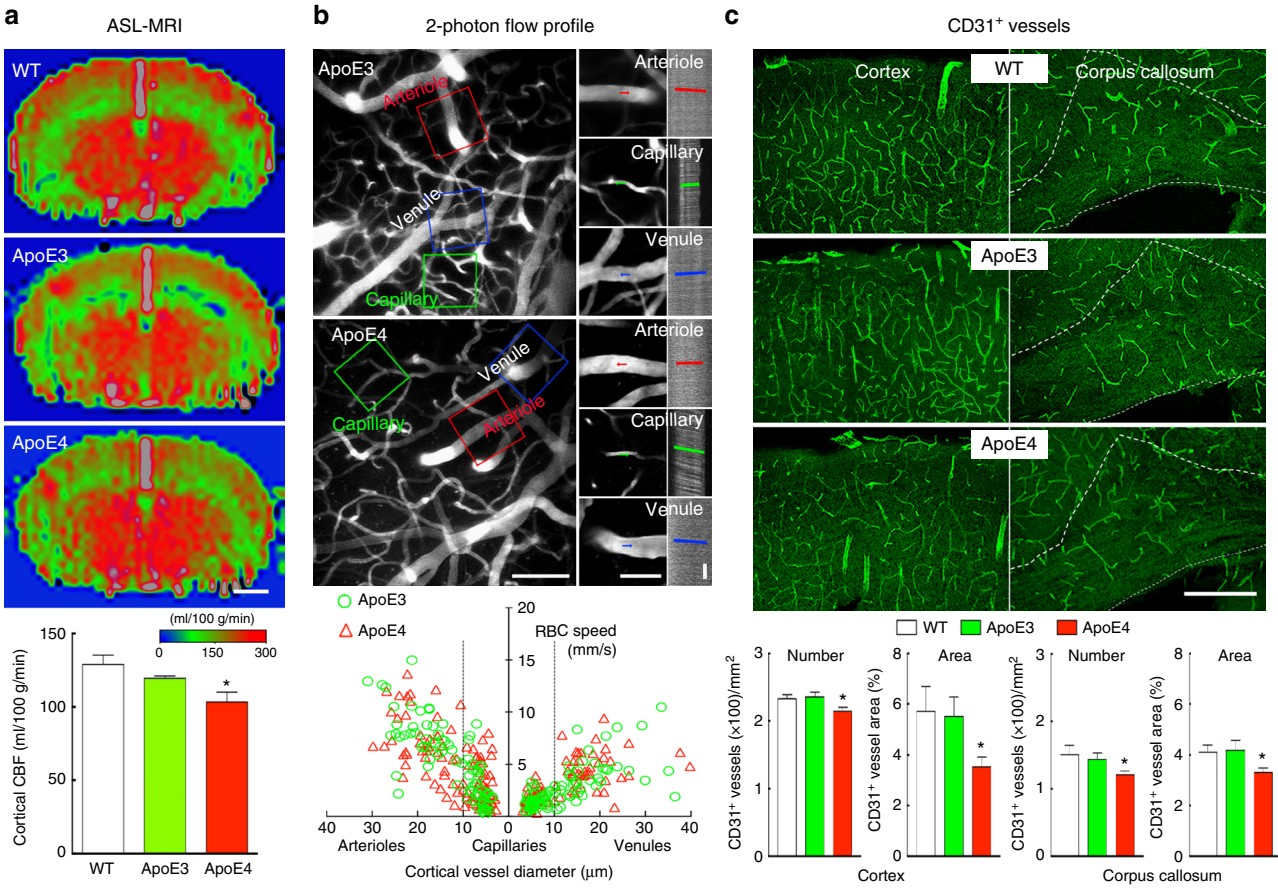

**Fig. 1** Reduced cortical CBF and vascular density in ApoE4-TR mice. **a** Neocortical CBF, assessed by ASL-MRI, is lower in ApoE4-TR mice, compared to WT and ApoE3 mice. Scale bar = 1 mm. **b** RBC speed in neocortical vessels (depth = 200 μm) in ApoE3-TR and ApoE4-TR mice assessed by 2-photon microscopy after injection of FITC-labeled dextran to label the vasculature. Top two images show the microvasculature in ApoE3 and ApoE4 mice. The colored rectangles indicate the location of the vessels (arterioles, capillaries, and venules) in which line scans were obtained to determine RBC speed (right panels). The bottom graph shows the RBC speed measured by line scans as function of the vessel diameter: arterioles (>10 μm) to the left, capillaries in the middle (≤10 μm; placed on arteriole or venule side depending on whether they were topologically closer to an arteriole or venule) and venules (>10 μm) to the right (ApoE3 n = 166 vessels in four mice, ApoE4 n = 197 vessels in four mice). Scale bars are 100 μm, 50 μm, 100 ms, respectively, for the large, small, and line scan images. **c** CD31+ microvessels are reduced in ApoE4 mice both in neocortex and CC. Scale bar = 250 μm. The ApoE3-TR mice did not differ from WT mice (p > 0.05). Data are expressed as means ± SEM. N = 5/group in **a** and **c**; *p < 0.05; one-way ANOVA and Tukey's test

(Fig. 2a–c; Supplementary Fig. 6), implicating ROS in the mechanisms of the neurovascular dysfunction.

**BCAS induces more marked CBF reductions in ApoE4-TR mice.** Considering their lower resting CBF and impaired neurovascular reactivity, we hypothesized that ApoE4-TR mice would be more susceptible to the hypoperfusion produced by BCAS[24]. Mice were equipped with bilateral LDF probes overlying the somatosensory cortex for chronic recording of CBF bilaterally (Fig. 3a). To reduce arterial diameter, 0.18 mm microcoils were placed around both common carotid arteries, and the corresponding reductions in cortical CBF were monitored at different time points (Fig. 3a, b; Supplementary Fig. 7A). In WT and ApoE3-TR mice, BCAS induced a reduction in CBF, which was greatest at 2 h (−40%) and remained suppressed 4 weeks later (−30%) (Fig. 3a, b). Using the resting CBF value provided by ASL-MRI (Fig. 1a) and the % reduction provided by LDF (Fig. 3a), we calculated in WT mice that cortical CBF was 77 ± 4 ml/100/min at 2 h and 99 ± 2 ml/100/min 4 weeks after BCAS (Supplementary Fig. 7B). However, in ApoE4-TR mice the reduction in CBF was more severe compared to WT and ApoE3-TR mice (p < 0.05)(Fig. 3a; Supplementary Fig. 7B). Using laser

speckle imaging (LSI), we confirmed that the reduction in CBF involves the entire neocortical mantle and is similar in magnitude and time course to that observed with LDF (Fig. 3b; Supplementary Fig. 7C). Therefore, the reduction in neocortical CBF produced by BCAS is more pronounced in ApoE4-TR mice.

**BCAS slows blood flow in CC more markedly in ApoE4-TR mice.** To determine whether BCAS reduces cerebral perfusion also in the WM of the CC of ApoE4-TR mice, we imaged RBC flow in CC microvessels before and 4 weeks after BCAS in the same mice using 3-photon microscopy[25]. The vasculature of the CC, ≈900 μm below the pial surface, was imaged and microvascular diameter and RBC speed were measured using line scanning[19,25] (Fig. 4a–e). At baseline, microvascular diameter did not differ between ApoE3-TR and ApoE4-TR mice, but RBC speed was slower in ApoE4-TR mice (Fig. 4f, g). Following BCAS, RBC speed decreased in both strains of mice, but the flow reduction was more severe in ApoE4-TR than in ApoE3-TR mice (Fig. 4g).

**BCAS damages WM and cognition more severly in ApoE4-TR mice.** To assess the impact of the more severe CC flow reduction

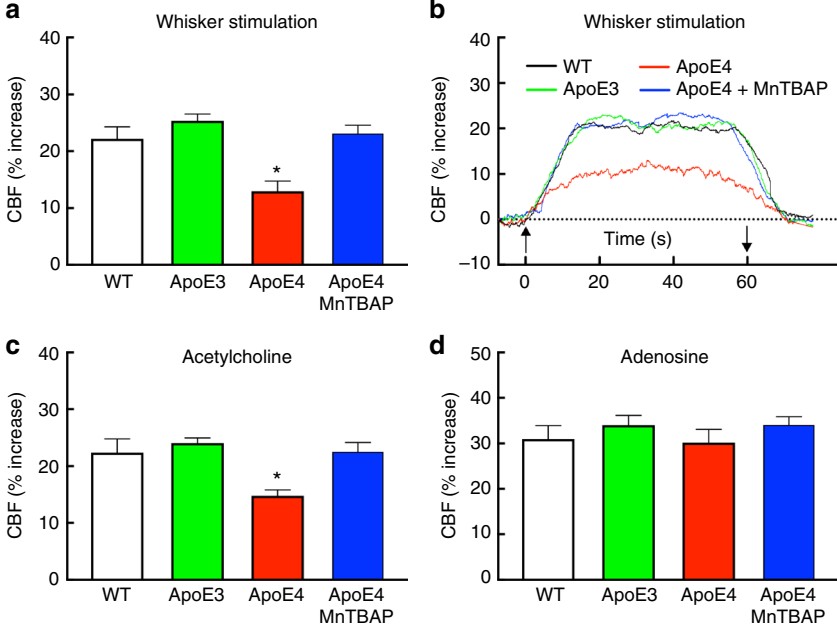

**Fig. 2** CBF dysfunction in ApoE4-TR mice is reversed by MnTBAP. The increase in CBF induced by whisker stimulation (**a**, **b**) or by neocortical application of acetylcholine (**c**), is attenuated in ApoE4-TR mice, but CBF response to neocortical application of adenosine (**d**) is not affected. Neocortical superfusion with the ROS scavenger MnTBAP restores neurovascular function in ApoE4-TR mice (**a–c**). Data are expressed as means ± SEM. $N = 5$/group; * $p < 0.05$ from WT or ApoE3, or ApoE4 + MnTBAP; one-way ANOVA and Tukey's test

in ApoE4-TR mice, we examined the degree of local hypoxia produced by BCAS in the CC using the hypoxyprobe. We found that the intensity of the hypoxia signal was more marked in ApoE4-TR than in ApoE3-TR and WT mice (Fig. 3c), suggesting that the greater reduction in WM CBF resulted in more severe WM hypoxia in ApoE4-TR mice. To determine if the more severe brain WM hypoperfusion and hypoxia in ApoE4-TR mice resulted in a worse cognitive outcome, we investigated the WM damage produced by BCAS and the attendant cognitive deficits in ApoE3-TR and ApoE4-TR mice. To this end, we examined indices of WM integrity 4 weeks after BCAS, when the damage is well developed[24]. BCAS resulted in demyelination of the CC as shown by the Klüver-Barrera stain and by the increased ratio between myelin basic protein (MBP) and the axonal marker SMI312 (Fig. 5a, b). Myelin-associated glycoprotein (MAG) was also reduced (Fig. 5c), while the microglial/macrophage marker Iba1 was increased in the CC, reflecting the microglial reaction to WM damage[26] (Supplementary Fig. 8). Consistent with increased BCAS-induced demyelination in ApoE4-TR mice, the oligodendrocyte marker Olig2 in the CC was reduced more in ApoE4-TR than in WT or ApoE3-TR mice (Supplementary Fig. 9). BCAS did not affect the reduction in pericyte vascular coverage observed at baseline in the CC of ApoE4-TR mice (Supplementary Fig. 2E–K). The integrity of the nodes of Ranvier was assessed by the spatial relationship between the paranodal contactin-associated protein Caspr and the sodium channel Na$_v$1.6, enriched at the nodes[27]. We found that BCAS increased the exposed nodal Nav1.6 channels, resulting in a looser association with Caspr and disruption of the nodal structure (Fig. 5d). BCAS also impaired the performance at the Y maze and novel object recognition tests, resulting in reduced arm alternation and novel object exploration time (Fig. 5e, f). All the indices of WM damage as well as cognitive performance were worse in ApoE4-TR, when compared to WT and ApoeE3 mice (Fig. 5e, f). No effect on locomotor activity was observed (Supplementary Fig. 10)

## Discussion

ApoE4, the major genetic risk factor for AD, has also emerged as a significant risk factor for SVD and WM lesions underlying vascular cognitive impairment[8–10]. To investigate the potential mechanisms of the effect we studied cerebrovascular structure and function in ApoE3-TR and ApoE4-TR mice. We found that ApoE4-TR mice have reduced resting CBF in the neocortex, an effect associated with vascular rarefaction. We also found an impairment of the major mechanisms linking cerebral perfusion to the energy demands of the brain, such as neurovascular coupling and endothelium-initiated microvascular responses[20]. These functional alterations were not due to a generalized vasomotor failure because the increase in CBF induced by the NO donor SNAP, the smooth muscle relaxant adenosine, or hypercapnia was not reduced. However, reduced cerebrovascular reactivity to hypercapnia has been observed by fMRI in ApoE4+ individuals[28]. Therefore, the ApoE4 genotype, in addition to reducing resting CBF, is also associated with a selective failure of fundamental homeostatic mechanisms matching cerebral perfusion with the metabolic requirements of the brain tissue.

To explore the impact of such neurovascular dysfunction on hypoxic-ischemic WM damage, we investigated whether the ApoE genotype influences the WM damage produced by cerebral hypoperfusion. There are several models of WM injury mimicking SVD, each with strengths and weaknesses[29]. BCAS has desirable features well suited for our purpose: (a) it leads to diffuse WM damage in the CC, a watershed region like the human subcortical WM; (b) the WM lesions are produced by hypoperfusion as in human WM disease;[1,5,30] (c) with 0.18 mm microcoils the lesions are restricted to the WM of the CC, sparing the optic nerve and the hippocampus;[24] (d) it avoids confounding effects of pharmacological models (e.g., endothelin-1), in which off-target effects cannot be ruled out;[29] and (e) it leads to quantifiable cognitive abnormalities attributed to damage of WM tracts connecting the hippocampus to other brain regions involved in cognition[24,31]. Using diverse approaches to monitor

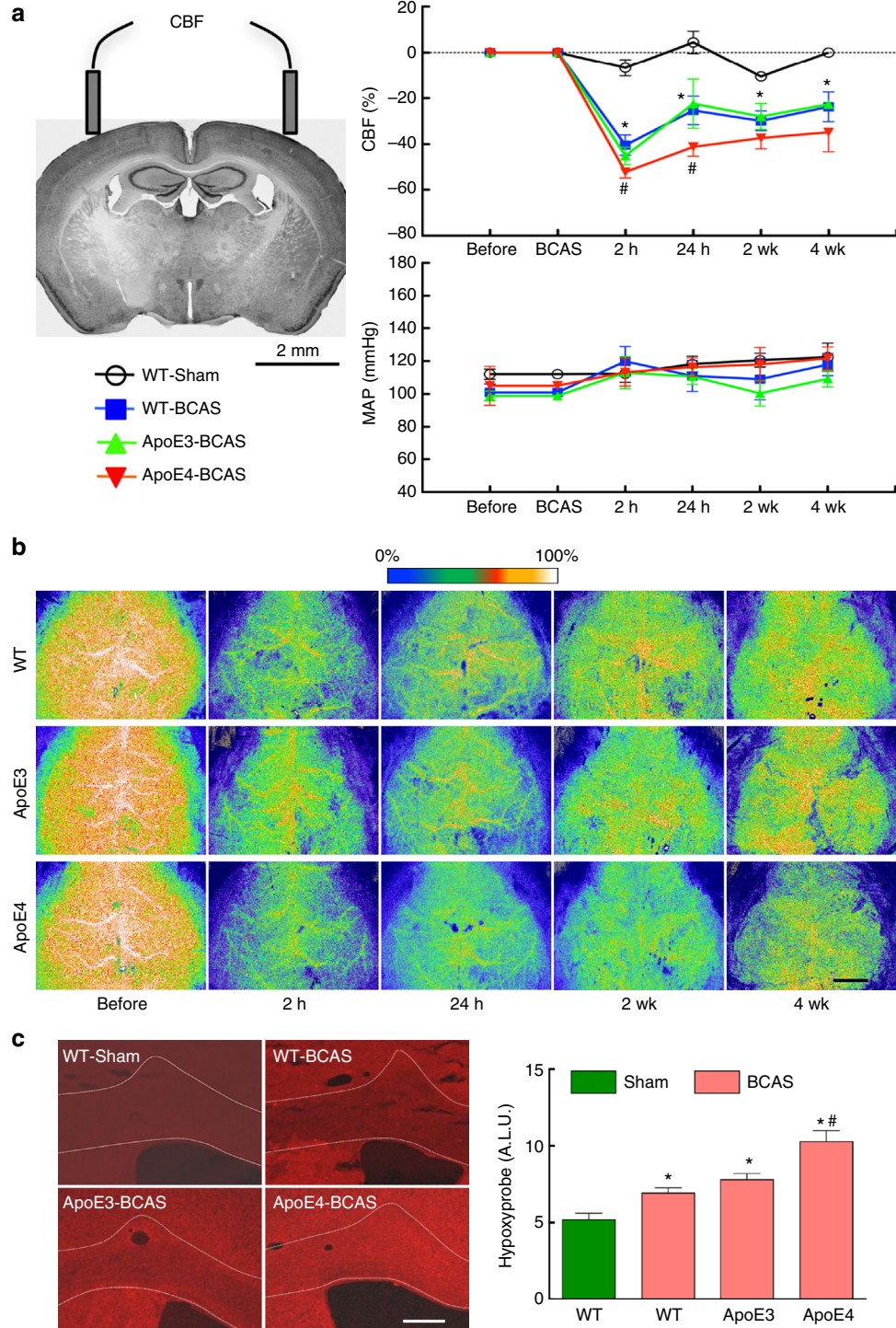

**Fig. 3** Worse cortex CBF and CC hypoxia in ApoE4-TR mice after BCAS. **a** Neocortical CBF and mean arterial pressure (MAP) recorded at the indicated times over 4 weeks before or after sham surgery (WT) or BCAS in WT (WT-Sham or -BCAS), ApoE3 (ApoE3-BCAS), and ApoE4 (ApoE4-BCAS) mice. Right and left CBF values were not different and were averaged. The CBF reduction induced by BCAS is more marked in ApoE4-TR mice. **b** Representative laser speckle images of CBF before or after BCAS in WT, ApoE3-TR, and ApoE4-TR mice, confirming the greater CBF reduction in ApoE4-TR mice (see Supplementary Fig. 7C for quantification). Scale bar = 2 mm. **c** The intensity of the hypoxyprobe signal, a marker of tissue hypoxia, is more pronounced in the CC of ApoE4-TR mice, compared to WT or ApoE3-TR mice. Data are expressed as means ± SEM. Scale bar = 250 μm. $N = 5$/group; * $p < 0.05$ from WT-sham; # $p < 0.05$ from WT and ApoE3-BCAS; one-way ANOVA and Tukey's test. ALU arbitrary fluorescent intensity units

microvascular perfusion, we demonstrated that BCAS reduces baseline CBF in cerebral cortex assessed by ASL-MRI, as well as in the microvasculature of the CC, assessed by 3-photon microscopy. The flow reduction was more marked in ApoE4-TR mice and was associated with more severe WM hypoxia and more

extensive WM damage, involving both myelin and nodal-paranodal structures. At the same time, ApoE4-TR mice exhibited more severe cognitive impairment than ApoE3-TR or WT mice. These observations collectively indicate that ApoE4 is associated with neurovascular dysfunction, leading to increased

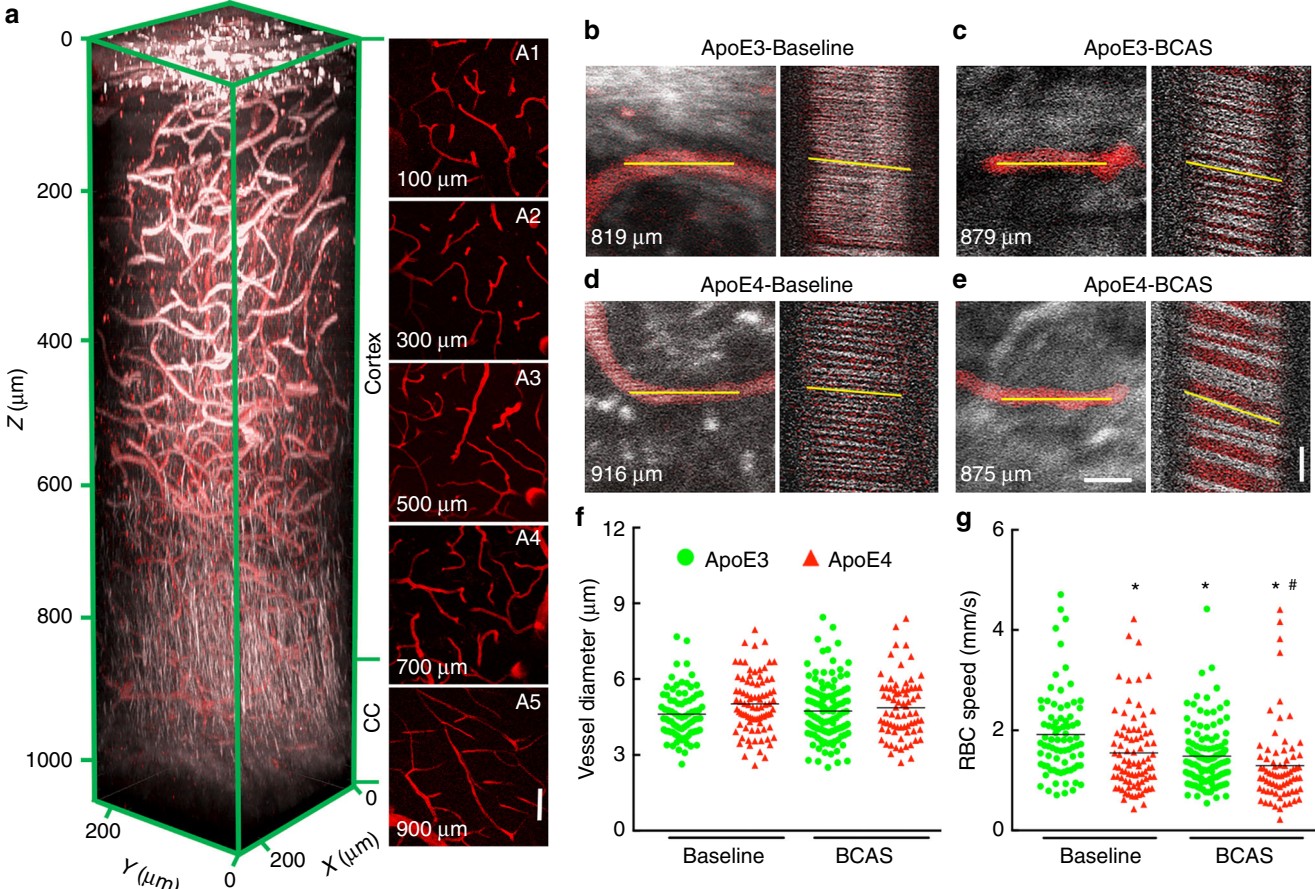

**Fig. 4** Reduced CC flow in ApoE4-TR mice before and after BCAS. **a** Rendering of in vivo 3-photon excited fluorescence/third harmonic generation (3PEF/THG) image stack that spans from the cortical surface to the white matter (WM) of the corpus callosum (CC) in ApoE3-TR mice. The vasculature was visualized by the THG signal (white), and was also labeled with FITC-dextran (red). The images to the right (A1–A5) show 40-μm thick maximal microvascular projections, centered at the indicated depths. Scale bar in A5 = 50 μm. **b–e** Projections (20 μm thick) of 3PEF/THG image stacks and line scans for capillaries in the CC, taken at the indicated depths from the cortical surface, of ApoE3 and ApoE4 mice at baseline and at one month after BCAS. Scale bars in E are 10 μm and 50 ms, respectively. **f, g** Plot of diameter (**f**) and RBC flow speed (**g**) in individual CC capillaries at the depth between 800–1000 μm in the same mice at baseline and 4 weeks after BCAS (ApoE3 baseline $n = 80$ in four mice, ApoE4 baseline $n = 85$ in four mice, ApoE3 BCAS $n = 118$ in four mice, ApoE4 BCAS $n = 73$ in three mice). Data are expressed as means ± SEM. * $p < 0.05$ from ApoE3-Baseline; # $p < 0.05$ from ApoE3-BCAS and ApoE4-Baseline; Kruskal-Wallis test

susceptibility to hypoperfusion-induced WM damage and cognitive impairment, and provide a potential mechanism for the increased WM damage in individuals carrying the ApoE4 gene in the absence of other vascular risk factors.

Previous studies reported a reduction in resting CBF in ApoE4-TR mice[18,32], but the microvascular bases of the effect remained unclear. Using 2-photon microscopy in carefully mapped vascular networks, we found that in ApoE4-TR mice baseline microvascular RBC flux and diameter are not reduced in neocortical arterioles, capillaries, and venules, but that vascular density is reduced. Therefore, the reduction in resting CBF is most likely due to a reduction in vascular density and not to reduced flow in individual microvessels. However, we cannot rule out the possibility that a reduction in cerebral metabolism, tightly coupled to CBF, could also contribute[32]. The deleterious effect of the reduction in baseline flow is compounded by the failure of critical homeostatic mechanisms matching the delivery of CBF to the metabolic needs of the tissue. In this regard, it is of interest that both the increase in CBF produced by neural activity and by the endothelium-dependent agonists ACh and BK are impaired in ApoE4-TR mice. Functional hyperemia is mediated by multiple vasoactive agents released by neurons and astrocytes, acting to relax the local microvasculature[20], whereas ACh and BK induced vascular relaxation through distinct mediators, respectively NO and cyclooxygenase-1 reaction products[33,34]. Therefore, the data point to a global disruption of vasoregulatory mechanisms involving both the endothelium and neurovascular pathways mediating functional hyperemia. Since, oxidative stress occurs in SVD in humans[22] and in animal models of neurovascular dysfunction[35,36], we asked whether ROS were also involved in the cerebrovascular effects of ApoE4. We found that the ROS scavenger MnTBAP is able to counteract in full the disturbance in cerebrovascular regulation. This finding confirms that the neurovascular dysfunction is not the result of irretrievable vascular damage. Furthermore, the observation that resting CBF, unlike the regulatory dysfunction, was not ameliorated by ROS scavenging suggests that the reduced baseline flow is not a consequence of the neurovascular dysfunction, which is consistent with being due to vascular rarefaction instead.

The cellular bases of the neurovascular dysfunction remain to be elucidated. Elegant studies focusing on the BBB have suggested that pericyte loss in 9-month-old ApoE4-TR mice may result in activation of the cyclophlin-A-NFκB-metalloprotease-9 pathway, leading to disruption of BBB integrity[18]. A transient reduction in

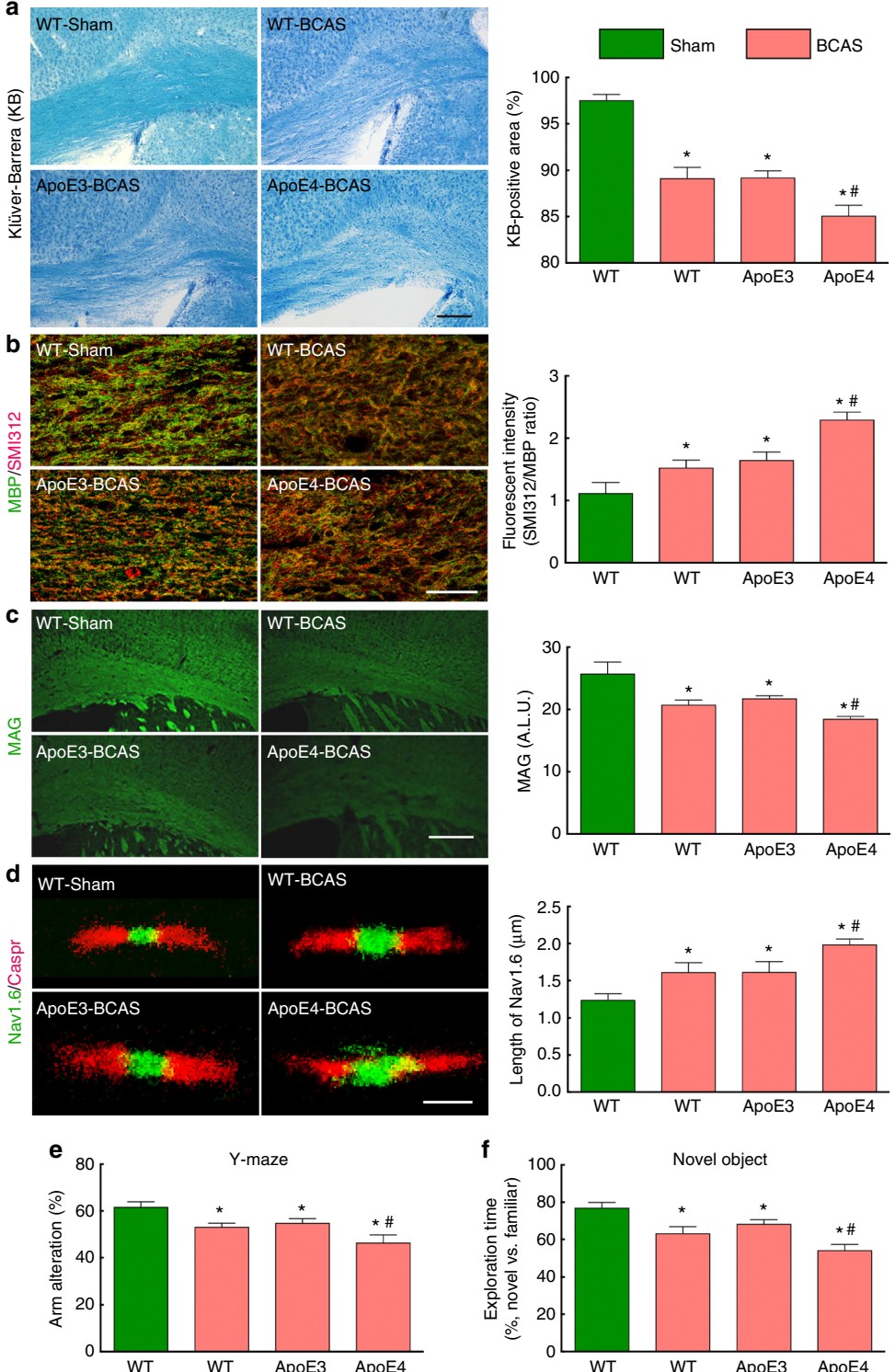

**Fig. 5** Worse CC injury and cognition in ApoE4-TR mice after BCAS. **a** Klüver-Barrera (KB) stain of the CC in sham-operated WT mice and in WT, ApoE3-TR, and ApoE4-TR mice after BCAS. In **a**, as in **b**, **c**, **d**, related quantification shown on the right. Scale bar = 250 μm. **b** Double-label immunohistochemistry with myelin basic protein (MBP) and the pan-axonal neurofilament marker SMI312 in the same groups of mice. Scale bar = 25 μm. **c** Myelin-associated glycoprotein (MAG) immunohistochemistry in the same groups of mice. ALU: arbitrary fluorescent intensity units. Scale bar = 250 μm. **d** Double-label immunohistochemistry for the nodal Nav1.6 channels and the paranodal protein Caspr in the same groups of mice. Scale bar = 1.5 μm. **e, f** ApoE4-TR mice have a more impaired arm alternation at the Y-maze test (**e**) and worse performance at the novel object recognition test (**f**). Data are expressed as means ± SEM. $N = 5$/group in **a**–**d**; $N = 10$/group in E–F; * $p < 0.05$ from WT-Sham; # $p < 0.05$ from WT- or ApoE3-BCAS; one-way ANOVA and Tukey's test

pericyte coverage and BBB dysfunction has also been reported 3 days after BCAS, but, as also shown in the present study, pericyte coverage is re-established at 28 days[37]. Therefore, our data that neurovascular coupling and endothelium-dependent responses are altered at baseline and are completely reversible by short-term ROS scavenging argue against pericyte loss being directly responsible for the neurovascular dysfunction. However, considering their critical role in neurovascular homeostasis and WM integrity[38], pericytes could contribute to the WM damage. For example, pericyte induced loss of BBB integrity could promote fibrinogen entry into the brain, resulting in oligodendrocyte damage and demyelination[5,39]. This hypothesis, in addition to hypoperfusion, would also implicate BBB disruption in the mechanisms of WM damage, but it needs to be tested in future studies. Similarly, the cellular sources of oxidative stress in ApoR4-TR has not been identified. Interestingly, ApoE4 macrophages generate more ROS than those expressing ApoE2 or ApoE3[23], but it remains to be established whether ApoE4 leads to greater ROS production in vascular and perivascular cells.

A unique aspect of our study is the use of 3-photon microscopy to explore microvascular perfusion in the WM of the CC in vivo, a task unattainable at the imaging depth afforded by 2-photon imaging. Using this technology, we demonstrated that WM microvascular flow is highly susceptible to reduction in cortical CBF, linking the WM damage produced by BCAS to the hypoxia caused by reduced cerebral perfusion. By imaging the CC microcirculation we discovered that the reduction in flow in ApoE4-TR mice is not the result of vascular compression or tissue edema[30], since the microvascular diameter is not reduced. Rather, we observed a reduction in RBC flow, which, in turn, led to reduced $O_2$ delivery and WM hypoxia. By combining these approaches, we established that in ApoE4-TR mice BCAS induces a more severe reduction in RBC speed and hypoxia, thereby providing insight into the mechanisms of the enhanced WM damage observed in these mice. It is also of interest that in naive ApoE4-TR mice baseline RBC speed was reduced in the CC, but not in the cortex. This difference remains unexplained, but it may be related to the precarious blood supply of the CC, which is vascularized by separate arterial territories: penetrating arterioles from the pial circulation and arterioles arising from the initial segment the middle cerebral artery and traveling though the striatum[5]. In this regard, our finding that in ApoE4-TR mice the reduction in CC microvascular flow is compounded by a reduction in microvascular density and pericyte coverage provides further insight into the bases of the greater vulnerability of the CC to hypoperfusion in these mice.

The findings shed light into the association of ApoE4 genotype, SVD and WM damage[11]. ApoE4 could be involved both in the pathogenesis of SVD as well as in the resulting WM damage. On the one hand, ApoE4-related oxidative stress and endothelial dysfunction may promote small vessels atherosclerosis and lipohyalinosis, pathological hallmarks of SVD[30,40]. On the other hand, reduced baseline CBF and neurovascular dysfunction afforded by ApoE4 may exacerbate hypoperfusion and hypoxia in the highly vulnerable subcortical WM. The combined effect of these alterations is likely to act in concert to promote hypoxic-ischemic WM damage leading to cognitive impairment.

In conclusion, we have demonstrated that the ApoE4 genotype is associated with reduced resting cerebral perfusion, coupled with a marked impairment of the major factors responsible for maintaining an adequate blood supply to the working brain. Furthermore, in conditions of cerebral hypoperfusion, ApoE4-TR mice develop more severe WM damage and cognitive impairment. The findings, collectively, provide insights into the mechanistic bases for the increased susceptibility to SVD and WM lesions in ApoE4 carriers, and highlight the prominent role

that genetic risk factors play in microvascular function and susceptibility to WM injury.

## Methods

**Mice**. All procedures were approved by the Institutional Animal Care and Use Committee of Weill Cornell Medicine and performed according to the ARRIVE guidelines[41], and, whenever appropriate, in a blinded fashion. Experiments were performed in homozygous ApoE3-TR and ApoE4-TR mice on a C57BL/6 genetic background[42,43]. ApoE2-TR mice were not used as controls since they develop atherosclerosis on a normal diet[44] and human ApoE2 may promote brain hemorrhage[45]. C57BL/6 mice were used as wild-type (WT) controls. All mice were males and aged 3–4 months.

**Absolute CBF measurement by arterial spin labeling MRI**. Imaging was performed on a 7.0 Tesla 70/30 Bruker Biospec small animal MRI system[46]. The animals were anesthetized with isoflurane (1–2%) and placed in the MRI. A volume coil was used for transmission and a surface coil for reception. Anatomical localizer images were acquired to find the transversal slice corresponding to the somatosensory cortex (from −0.7 to 0.38 from Bregma)[47]. This position was used for subsequent ASL imaging, which was based on a FAIR-RARE pulse sequence labeling the inflowing blood by global inversion of the equilibrium magnetization. Three averages of one axial slice were acquired with a field of view of 15 × 15 mm, spatial resolution of 0.234 × 0.234 × 2 mm³, echo time TE of 5.368 ms, effective TE of 26.84 ms, repeat time TR of 10 s, and a RARE factor of 36. For computation of CBF (ml/100 g/min), the Bruker ASL perfusion processing macro was used. Turbo-RARE anatomical images were acquired with the same field-of-view and orientation as the ASL images (resolution = 0.078 × 0.078 × 1 mm³, TE = 48 ms, TR = 2200 ms, and a RARE factor of 10). The total scan time (ASL + anatomical image) was approximately 40 min.

**Bilateral common carotid artery stenosis**. Mice were anesthetized with isoflurane (1–2%) in a mixture of oxygen-nitrogen with rectal temperature maintained at 37 °C[24,31]. Both common carotid arteries were dissected thorough a midline incision and microcoils (internal diameter: 0.18 mm; Sawane Spring, Japan) were placed around the arteries[24,31], (Supplementary Fig. 7A). Sham-treated mice underwent the same surgical procedure with no placement of microcoils.

**Monitoring of CBF**. CBF response to neural activity and to endothelium-dependent and independent agonists: Anesthesia was induced with isoflurane (1–2%) and maintained with urethane (750 mg/kg; i.p.) and α-chloralose (50 mg/kg; i.p.). A femoral artery was cannulated for recording of arterial pressure and collection of blood samples[36,48,49]. A 2 × 2 mm opening was drilled in the parietal bone overlying the somatosensory cortex, the dura was removed, and the site was superfused with a modified Ringer solution (37 °C; pH 7.3–7.4)[48]. Relative CBF was continuously monitored at the site of superfusion with a LDF (Perimed). Arterial blood pressure, blood gases, and rectal temperature were monitored and controlled. CBF recordings were started after arterial pressure (MAP, 78–85 mmHg) and blood gases (pO₂, 120–140 mmHg; pO₂, 33–40 mmHg; pH, 7.3–7.4) were in a steady state[36,48,49]. For functional hyperemia, the whiskers were mechanically stimulated for 60 s and the associated increase in CBF recorded over the somatosensory cortex. To test endothelium-dependent responses, ACh (10 μM; Sigma) or BK (50 μM; Sigma) was superfused over the cranial window, and the resulting changes in CBF monitored. CBF responses to superfusion with SNAP (50 μM; Sigma) and adenosine (400 μM; Sigma) were also tested. The CBF response to hypercapnia (pCO₂: 50–60 mmHg) was tested by introducing $CO_2$ through the ventilator[36,49].

Chronic CBF recordings after BCAS: CBF was monitored with LDF or LSI. For LDF, mice were anesthetized with isoflurane (1.5–2%) in a mixture of oxygen and nitrogen. Two fiber-optic probes were permanently glued to the parietal bone of both hemispheres (3.5 mm lateral to bregma) (see Fig. 3a) and connected to a 2-channel laser-Doppler flowmeter (PF 5010, Perimed) for longitudinal CBF monitoring in the same mouse. Baseline CBF was obtained 30 min before BCAS. At the same time, pre-stenosis MAP was recorded through a femoral artery catheter. Then, BCAS was induced by placing microcoils. For CBF measurement with LSI (Omegazone; Omegawave), mice were anesthetized with 1–2% isoflurane, and the scalp was removed to expose the skull. The following day, mice were re-anesthetized and the exposed skull was illuminated by laser light (780 nm). The scattered light was filtered and detected by a CCD camera positioned over the skull. The raw speckle images were used to compute speckle contrast, which in the mouse neocortex reflects the velocity of moving RBCs up to a depth of ≈700 μm, but weighted more towards surface vessels[50]. Color-coded blood flow images were obtained in high-resolution mode (639 × 480 pixels; 1 image/s) and the sample frequency was 60 Hz. One CBF image was generated by averaging numbers obtained from 20 consecutive raw speckle images. The recordings were initiated after the CBF was stable, and five recordings of blood flow image were averaged. The CBF reduction induced by BCAS was calculated as a percentage of the pre-stenosis CBF value. For both LDF and LSI, CBF changes were recorded 2 hours, 4 hours, 2 weeks, and 4 weeks after BCAS.

**Multiphoton imaging**. Optical access to brain was achieved through a long-term glass-covered cranial window implanted between lambda and the bregma (8 wide x 4 mm long)[51]. Animals were anesthetized using isoflurane (1.5–2% in oxygen) and maintained at 37 ˚C rectal temperature. Bilateral craniotomies were performed over parietal cortex using a dental drill and sealed with a glass coverslip. All mice recovered at least 21 days before in vivo imaging and BCAS surgery.

Multiphoton excited fluorescent microscopy: Imaging was conducted using a custom multiphoton microscope. For 3-photon excited fluorescence (3PEF) imaging with 1300 nm laser light, we used an optical parametric amplifier (OPA) (Coherent, Opera-F) that was seeded by a diode-pumped femtosecond laser (40 μJ/pulse at 1 MHz; Coherent, Monaco), producing sub-100-fs pulses at 1-MHz repetition rate with up to 1-μJ energy. Dispersion for 1300-nm excitation was compensated with an SF11 prism pair. For imaging, the excitation laser was scanned with a line rate of ~1 kHz with galvanometric scanners. Images were acquired with an Olympus XLPlan N $25 \times 1.05$ NA objective. Signal was collected using custom detection optics using emission filters (center wavelength/bandwidth): 417/60 (third harmonic generation, THG) and 494/41 (fluorescein isothiocyanate, FITC), separated with a long-pass dichroic with a cutoff at 458 nm. For 2-photon excited fluorescence (2PEF) imaging, a Ti:Sapphire oscillator (Chameleon, Coherent) at 800-nm wavelength was used, and FITC emission was collected with the same filter. Image acquisition was controlled by ScanImage software[52].

Imaging cortical and subcortical blood flow: Cortical blood flow was measured using 2PEF imaging, while CC blood flow was characterized using 3PEF microscopy, all on a microscope that was equipped with both excitation laser sources. For cortical measurements using 2PEF imaging, the blood plasma was labeled with 2.5% w/v FITC 70 kDa conjugated dextran (Invitrogen) diluted in sterile saline and retro-orbitally injected (50 μl) before imaging. We took 3D image stacks and identified individual arterioles, capillaries, and venules (noting both spatial location and topological position in the vascular hierarchy) for measurement of vessel diameter and RBC speed using image stacks and lines cans[19]. RBC speed measurements were made at a depth of 200 μm in all groups of animals.

For 3PEF measurements of CC capillaries, we used the signal from FITC-dextran to identify vessels and measure vessel diameter. However, we used the endogenous THG signal from RBC[53,54] for the line scan measurements (Supplementary Fig. 11), since THG provides a higher contrast signal compared to the 3PEF of FITC-dextran, enabling higher fidelity measurement of blood flow speed[53,54]. The 3PEF imaging was performed between 800 and 1000 μm (typically 900 μm), depending on the depth of the CC relative to the pial surface. In addition to the depth (>800 μm), the localization to the CC was verified by: (a) the distinctive pattern of the THG signal arising from the WM, reflecting the more compact WM tracts of the CC, and (b) the linear pattern of microvessels and reduced vascular density indicated by the FITC signal (Fig. 4A5). All images were processed using ImageJ (National Institutes of Health), and blood cell speed was calculated by custom written MATLAB code[19,55].

**Detection of hypoxia by the hypoxyprobe**. Hypoxia was detected using Hypoxyprobe™-1 kit (Hypoxyprobe Inc, Burlington, MA) according to the manufacturer's instruction. Four weeks after BCAS, mice were injected with Hypoxyprobe-1 (60 mg/Kg; i.p.). One hour later, anesthetized mice were trans-cardially perfused. Brains were removed, postfixed, and sectioned (16 μm) with a cryostat. Brain sections were incubated with a monoclonal anti-hypoxyprobe antibody (1:50) at room temperature for 2 h. Images were obtained with a confocal laser-scanning microscope (Leica SP5), quantified by ImageJ (NIH) and difference in stain intensity expressed as arbitrary luminescent unit.

**Evaluation of WM injury**. Four weeks after BCAS, anesthetized mice were perfused transcardially[36,49]. Brains were removed, postfixed overnight, and sectioned with cryostat (thickness of 12 μm) or vibratome (40 μm). For immunohistochemical evaluation of BCAS-induced WM injury the following antibodies were used with the appropriate secondary antibodies (Supplementary Table 1): anti-cluster of differentiation 31 (CD31) (rat; 1:30; BD Pharmingen), anti-glucose transporter 1 (Glut1) (rabbit; 1:500; Millipore Sigma), anti-myelin basic protein (MBP) (rat; 1:500; Millipore Sigma), the neurofilament anti-SMI312 (mouse;1:500, Covance), anti-myelin-associated glycoprotein (MAG) (10 μg/ml; Millipore Sigma)[56], anti-contactin-associated protein (Caspr) (mouse; 1:300, Millipore Sigma)[27], anti-voltage-gated sodium channel (Na$_v$) 1.6 (Na$_v$ 1.6) (rabbit; 1:200, Alomone)[27], anti-Ionized calcium binding adaptor molecule 1 (Iba1) (rabbit; 1:500; Wako), anti-oligodendrocyte transcription factor 2 (Olig2) (rabbit; 1:200; Millipore Sigma), or anti-synaptophysin (mouse; 1:100; R&D Systems). Images were obtained with a confocal laser-scanning microscope (Leica SP5) and analyzed using ImageJ. The Klüver-Barrera stain was performed using the Luxol Fast Blue Stain Kit (ScyTek Laboratory Inc.). Brains were harvested after transcardiac perfusion with PBS and 4% PFA, sectioned with a vibratome (thickness 40 μm), and the positive (blue stained) area in the CC was quantified by ImageJ.

**Cytoplasmic Ca$^{2+}$ in dissociated neocortical neurons**. Coronal cortical slices (350 μm in thickness) were cut from the brain of 3-month old WT, ApoE3-TR and ApoE4-TR mice and incubated aCSF containing Pronase 0.02% (w/v), thermolysin (0.02%) at 36 °C (Sigma Aldrich)[57]. The tissue was mechanically dissociated and incubated with the Ca$^{2+}$ indicator Fura-2/AM (20 μmol/l, Life Fisher)[58]. Fura-2/AM-loaded cells transferred to polyornithine-coated glass-bottom Petri dish (Warner Instruments, CT). Images were acquired on a Nikon 300 inverted microscope by using a fluorite oil-immersion lens (Nikon CF UV-F X40; N.A., 1.3)[58]. Fura-2/AM was alternately excited through narrow bandpass filters (340 and 380 nm). An intensified CCD camera (Retiga ExI) recorded the fluorescence emitted by the indicator (510 ± 4 nm). After background subtraction, fluorescence measurements were obtained from cell bodies of 3–5 neurons in randomly selected fields before and after perfusion of the cells with NMDA (40 μmol/l) for 20 min. Fluorescence ratios (340/380 nm) were calculated for each pixel by using a standard formula[59]. At the end of the experiment, the viability of the preparation was verified by testing the Ca$^{2+}$ response to the Ca$^{2+}$ ionophore A23187[58].

**Cognitive testing**. We elected to use the Y-maze and novel object recognition because these tests: (a) are sensitive to the cognitive effects of BCAS[35,60–63], (b) rely on WM tracts connecting the hippocampus to the cortex and other brain regions[64] and, as such, are appropriate in a model of WM damage, (c) rely on the spontaneous behavior of mice and (c) do not required aversive environments or starving of the mice[65–67].

Y-maze spontaneous alternation behavior: mice were placed into one of the arms of the maze (start arm) and allowed to explore only two of the three arms for 5 min (training trial). The closed arm was opened in the test trial, serving as the novel arm. After a 30-min interval between trials, the mice were returned to the same start arm and were allowed to explore all three arms for 5 min (test trial). Sessions were video recorded and analyzed using AnyMaze (San Diego Instruments) in a double-blinded fashion. Spontaneous alternation was evaluated by scoring the order of entries into each arm during the 5 min of the test trial. Spontaneous arm alternation (%) was defined as: number of arm alternations/(total number of arm visits-2) x 100.

Novel object recognition: the test was performed in two consecutive days. On day one, mice were placed in the center of an empty open box and allowed to explore for 5 min. The box was cleaned with 70% ethanol between trials. On day 2, the mice were placed back to an open box with two identical objects in the center and allowed to explore for 5 min. Thirty minutes later, mice were exposed again to a familiar and a novel object, and allowed to explore for 5 min. The exploring activity (facing, touching, or sniffing the object) was monitored and analyzed using AnyMaze in a double-blinded manner, and the percent of the time spent exploring the novel vs. familiar objects was calculated.

**Data analysis**. Statistical analysis was performed using GraphPad Prism (GraphPad Software, Inc). Histological and cerebrovascular analyses were conducted in a blinded fashion. Samples and animals were randomized by a random number generator (www.random.org). No animals were excluded. The number of mice required for assessing statistical significance of pre-specified effects was estimated by power analysis based on preliminary results and previous experience with the models used in the lab. Normality was tested using D'Agostino-Pearson normality test (GraPad Prism) before running appropriate statistical tests. Two-group comparisons were analyzed by the two-tailed $t$-test and multiple comparisons were evaluated by the analysis of variance and Tukey's test, after testing for equality of variance. The data in Fig. 4G were evaluated by the nonparametric Kruskal-Wallis test. Differences were considered statistically significant for probability values less than 0.05. Data are expressed as means ± SEM.

## Data availability

The authors declare that the data supporting the findings of this study are available within the article and Supplementary Information Files or available from the authors upon reasonable request.

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

## Acknowledgements

This study is supported by grants (R01-NS100447, C.I.; R01-NS097805, L.P.) from National Institute on Aging (NIA) and National Institute of Neurological Disorders and Stroke (NINDS). Y.H. is supported by fellowships from Japan Heart Foundation/Bayer Research Grant Abroad, The Uehara Memorial Foundation Research Fellowship, and Japan Society for the Promotion of Science Overseas Research Fellowships. Support from the Feil Family Foundation is gratefully acknowledged.

## Author contributions

K.K. and Y.H. conducted the experiments and performed the data analysis; S.J.A. conducted the two- and three-photon microscopy experiments; I.B., K.U., and A.C. contributed to the histology experiments; G.W. and A.H performed the $Ca^{2+}$ imaging experiments; H.U.V. contributed to the ASL- and T2-MRI experiments and data analysis; C.S. supervised and analyzed the experiments involving the two- and three-photon experiments; L.Z., S.M.P., L.P., and C.I. supervised the research and wrote the manuscript.

## Additional information

**Competing interests:** The authors declare no competing interests.

