## [Peer Review File · Nature Communications]

Reviewers' comments:

Reviewer #1 (Remarks to the Author):

The investigators report that mice with human ApoE4 have reduced cortical blood flow compared to ApoE3-TR mice with a reduction in vascular density accounting for the flow decrease. Functional hyperemia and endothelial mediated hyperemia were attenuated in E4 mice and could be fully restored by a ROS scavenger. The investigators further exposed control, E3, and E4 mice to bilateral carotid artery stenosis. This BCAO exacerbated damage to the white matter (corpus callosum) and produced further cognitive dysfunction in E4 mice. The white matter damage was linked to reductions of microvascular flow (RBC velocity) resulting in local hypoxia. The authors conclude, "The findings unveil a previous unrecognized ApoE4-dependent failure of neurovascular regulation, sufficient to induce white matter hypoxia and damage. Such alterations may be responsible for the increased susceptibility to hypoxic-ischemic lesions in the subcortical white matter of individuals carrying the ApoE4 allele".

Abstract: "ApoE3 have reduced neocortical cerebral blood flow compared to ApoE3-TR mice, an effect due to reduced vascular density rather than slowing of microvascular flow". Really should read "slowing of microvascular RBC velocity". Address the relationship between velocity and flow.

Quantitative CBF was for the caudate and white matter was not provided in the manuscript but should be available from the ASL-MRI measurement. Inclusion of this information is important.

Line 97 "We focused on the same cortical area where CBF was measured by ASL-MRI ..." Exactly where was the cortical area.

Lines 108 and 109 "Therefore, vascular density and resting cerebral perfusion are reduced in ApoE4-TR compared to ApoE3-TR mice." This should read vascular density is reduced and resistance is increase in ApoE4-TR compared to ApoE3-TR mice. In fact, perfusion pressure would have to remain the same if vessel diameters and velocity were the same in the two groups.

Figures 3A and S4B provided % change in CBF following BCAS. It would be very informative to provide real numbers to these % changes after BCAS. Cortical blood flow during the resting state was quantitatively measured as ~115 and ~100 ml/100g/min (fig 1). For example, at 24 hours a ~30% reduction in CBF occurred in ApoE3 and a ~45% reduction occurred in CBF in ApoE4. This would provide absolute flows of approximately 70 and 55 ml/100g/min in CBF for ApoE3 and ApoE4 mice respectively.

ApoE3 flows are: $\sim 115 \text{ ml/100g/min} \times 0.7$ (30% reduction) = 80 ml/100g/min

ApoE4 flows are: $\sim 100 \text{ ml/100g/min} \times 0.55$ (45% reduction) = 55 ml/100g/min

This difference in absolute flows is impressive and should be reported. Perhaps a graph of absolute flows over the 4 week period could accompany figure 3A (either text or supplement) or figure S4B.

Reviewer #2 (Remarks to the Author):

APOE4 is the strongest and most highly replicated genetic risk factor for sporadic Alzheimer's disease (AD). APOE4 increases blood-brain barrier (BBB) damage, cerebral amyloid angiopathy and fibrinogen deposits in the human brain, and has been implicated in increasing oxidative stress in brain by increasing levels of reactive oxygen species (ROS). Furthermore, APOE4 carriers have accelerated pericyte degeneration and BBB breakdown compared to APOE3 carriers, likely dependent upon the cyclophilin A/matrix metalloproteinase-9 pathway. In this manuscript, the authors find that TR-ApoE4 mice have reduced basal neocortical cerebral blood flow (CBF) due to

reduced vascular density. CBF change in response to whisker stimulus or application of endothelial-specific vasodilators acetylcholine and bradykinin was also reduced in TR-ApoE4 mice, which could be rescued by application of ROS scavengers to the cortex. Next, the authors induced bilateral common carotid artery stenosis in TR-APOE mice and found increased white matter damage and reduced microvascular flow causing local hypoxia in TR-ApoE4 compared to TR-ApoE3 mice. This is a very interesting paper with cutting-edge methodologies and with potential great impact on better understanding complex cerebrovascular effects of apoE4. Here, we offer several comments that may significantly strengthen this manuscript.

Major comments:

- As the authors acknowledge in the discussion, the main weakness of this paper is the lack of mechanism of how white matter damage and reduced microvascular density is occurring in TR-ApoE4 mice. Previous studies have shown reduced pericyte coverage in APOE4 human carriers (Lancet Neurol. 2011 Mar;10(3):241-52.; J Cereb Blood Flow Metab. 2016 Jan;36(1):216-27) and TR-ApoE4 mice (Nature. 2012 May 16;485(7399):512-6). Furthermore, the loss of pericytes leads to CBF reductions and white matter dysfunction (Nat Med. 2018 Mar;24(3):326-337). Therefore, the authors are highly encouraged to measure pericyte coverage in the white matter of TR-ApoE mice, and perhaps make the link to what has been known in the literature.
- The authors state on page 9, top paragraph, that, "with 0.18mm microcoils the lesions are restricted to the WM of the CC, sparing the optic nerve and the hippocampus, which is a critical requirement for cognitive testing 32." However, the authors performed hippocampal-dependent learning tests (Y-maze and novel object recognition) and found deficits. Therefore the authors' statement should be revised with this in mind. Furthermore, it might be informative to include behavioral tasks more associated with white matter deficits such as radial arm maze (Stroke. 2007 Oct;38(10):2826-32) and complex running wheel (Nat Med. 2018 Mar;24(3):326-337).
- If possible, the authors should consider adding quantification of white matter hyperintensities using Fazekas scale (Neurology. 1993 Sep;43(9):1683-9).
- Did the authors consider measuring changes to oligodendrocytes that may lead to white matter loss? The link pericyte loss, BBB WM breakdown, fibrinogen, oligodendrocytes cell deaths has been established. Perhaps, please consider it for discussion.
- Of relevance to the paper, impaired cerebral blood vessel reactivity to CO₂ has been shown in the hippocampus of young, non-demented APOE4 carriers by reduced CBF velocity in the middle cerebral artery using transcranial Doppler (J. Am. Geriatr. Soc. 63, 276–281 2015) and by BOLD-fMRI in response to a memory task (Alzheimers Dement. 11, 648-657.e1 2015). The authors may consider discussing these observations.
- To provide a complete picture to the readers, the authors perhaps may consider briefly mentioning in the Introduction that APOE isoforms differentially affect the binding and trans-vascular clearance of A β , with impaired clearance in APOE4 carriers. (J Clin Invest. 2008 Dec;118(12):4002-13).
- The three-photon vascular diameter and RBC velocity data are interesting. It would greatly strengthen the manuscript to show that the authors are indeed imaging from the CC, as it is difficult to tell from the 3D volume representation in figure 4A. It would also be helpful to better clarify what depth the authors consider the top of the CC, since in figure 4A the authors mark depths of approx. 870-1050um as CC, but indicate in the figure 4 caption that CC RBC velocity measurements were made between 700-1000um.
- For the two-photon measurements of RBC velocity (presented in Fig. 1), it would be helpful to clarify over what range of depths the measurements were made, and if the range of depths were similar for all the animals studied.

Minor comments

- For bar graphs with multiple comparisons (more than 2 bars), it would be helpful for the reader if the authors add lines/indicators to distinguish which group comparisons are significant. See for example Figure 1A, is ApoE4 significant compared to either WT, ApoE3 or both? Adding this visual cue would be helpful.
- The authors should consider including a table of all antibodies used in the supplementary

materials.

- It would also be helpful to provide more details of the imaging parameters, methods, and depth ranges used for two- and three-photon imaging in the methods. At the least, it would be helpful to provide an explanation of what the "custom written MATLAB code" for measurement of RBC velocity does. Is this a variant of one of the published velocity measurement softwares? Similarly, it would be beneficial to provide more details of the vessel diameter measurement methodology.
- In figure 1B, it would be helpful to indicate the depth over which the vascular projection images were made. Also, in the plot of velocity vs. diameter, it would be helpful to provide some kind of lines or marks to better indicate the capillary diameter range. Since the capillary diameter range is <10um and venule range is >11 um, are the venule-side vessels between 10-11um diameter shown on the plot?
- Please provide an example of a THG channel image used for RBC velocity measurements separate from the dextran dye. The merged images in Fig. 4 are colorful, but make it more difficult to distinguish the THG signal.
- In figure 3B, the color scale bar does not match the color scale used in the images.
- Figure 5B would be more convincing with better quality representative images.
- In figure 4G, the data does not look normally distributed. Was it tested for normality before running statistics?
- Line 223: "vascular density" may be a more descriptive term than "vascularity."
- The manuscript should be checked carefully for typos, e.g. "SMI32," "ApoE4=TR," "box plot," incorrect supplemental figure reference in Methods.

Reviewer #3 (Remarks to the Author):

While apoE4 is an important risk factor for both Alzheimers disease and vascular dementia most of the apoE4 in vivo mice model studies to date focus on the neuronal and inflammatory effects of apoE4. This missing link is provided by the present study which focuses on the vascular effects of apoE4 in targeted replacement mice. The results obtained first revealed that under resting conditions the cerebral blood flow of the apoE4 mice is reduced relative to that of the apoE3 mice and that this is associated with a reduction in the number and area of the microvessels. The study than focused onthe neurovascular unit and the coupling between neuronal excitation and cerebral blood flow. This revealed that, like under basal conditions, the increase in cerebral blood flow in the somatosensory cortex following whisker stimulation was markedly reduced in the apoE4 mice. Furthermore, stenosis induces cerebral hypoperfusion in apoE4 mice which is associated with more severe white matter hypoxia ,myelin disruption and cognitive impairments.

The present experimental design in which the authors look at the vasculature at several complimentary levels including basal and stimulated conditions as well as under stress challenges is comprehensive and impressive and adds significantly to the impact of the observatio This is further fortified by the consistency of the observations regarding the vascular effects of apoE4 under the different conditions.

The interplay between the neuronal and vascular effects of apoE4 is still an open question regarding which the presently studied experimental system could shed some light. For example, it would be of interest to assess the extent to which the whisker barrel stimulation vascular impairments in the apoE4 mice are associated with corresponding cortical neuronal impairments. This is of particular interest as the response of the vasculature to pharmacological treatments (eg the NO donor SNAP or the smooth muscle relaxant adenosine) are not affected by apoE4. In addition, the authors should include immunohistochemical data of the effects of apoE4 on neuronal parameters such as synaptophysin and if possible on microglial parameters such as iba. Since previous studies have focused primarily on the neuronal and inflammatory effects of apoE4 we feel that the impact and significance of this study will be markedly increased by such additions Please add the age and gender of the mice. These technical points are important in order to establish how early the observed vascular pathology evolves and in view of the fact that in man and mice females are more susceptible than male to apoE4.

We wish to thank the reviewers for their careful reading of the manuscript and helpful suggestions, which have improved the quality of the presentation.

Reviewer #1 (Remarks to the Author):

The investigators report that mice with human ApoE4 have reduced cortical blood flow compared to ApoE3-TR mice with a reduction in vascular density accounting for the flow decrease. Functional hyperemia and endothelial mediated hyperemia were attenuated in E4 mice and could be fully restored by a ROS scavenger. The investigators further exposed control, E3, and E4 mice to bilateral carotid artery stenosis. This BCAO exacerbated damage to the white matter (corpus callosum) and produced further cognitive dysfunction in E4 mice. The white matter damage was linked to reductions of microvascular flow (RBC velocity) resulting in local hypoxia. The authors conclude, "The findings unveil a previous unrecognized ApoE4-dependent failure of neurovascular regulation, sufficient to induce white matter hypoxia and damage. Such alterations may be responsible for the increased susceptibility to hypoxic-ischemic lesions in the subcortical white matter of individuals carrying the ApoE4 allele".

Response: we thank the reviewer for the concise summary of our findings.

Abstract: "ApoE3 have reduced neocortical cerebral blood flow compared to ApoE3-TR mice, an effect due to reduced vascular density rather than slowing of microvascular flow". Really should read "slowing of microvascular RBC velocity". Address the relationship between velocity and flow.

Response: This point is well taken, and we have modified the abstract accordingly (page 2, line 42).

Quantitative CBF was for the caudate and white matter was not provided in the manuscript but should be available from the ASL-MRI measurement. Inclusion of this information is important.

Response: We could not reliably measure CBF in the corpus callosum due to the small size of this structure and to the limited resolution of conventional ASL-MRI in the mouse brain [see Shen and Duong (2016) ¹]. Therefore, we used 3-photon microscopy to assess red blood cell flow in the corpus callosum (Fig. 4). However, we were able to measure CBF in the caudate nucleus and found a 14±4% reduction in ApoE4-TR vs. WT mice. These new data were included in a Supplementary Fig. 1A and described in the text (page 4, lines 96-98).

Line 97 "We focused on the same cortical area where CBF was measured by ASL-MRI ..." Exactly where was the cortical area.

Response: We have specified that the cortical area studied was the somatosensory cortex (Page 4, line 100).

Lines 108 and 109 "Therefore, vascular density and resting cerebral perfusion are reduced in ApoE4-TR compared to ApoE3-TR mice." This should read vascular density is reduced and resistance is increase in ApoE4-TR compared to ApoE3-TR mice. In fact, perfusion pressure would have to remain the same if vessel diameters and velocity were the same in the two groups.

Response: Thank you. We have modified the statement to read "The vascular density and resting cerebral perfusion are reduced in ApoE4-TR mice, which, in the absence of perfusion pressure changes, suggests an increase in cerebrovascular resistance compared to ApoE3-TR mice" (page 5, line 115-117).

Figures 3A and S4B provided % change in CBF following BCAS. It would be very informative to provide real numbers to these % changes after BCAS. Cortical blood flow during the resting

state was quantitatively measured as ~115 and ~100 ml/100g/min (fig 1). For example, at 24 hours a ~30% reduction in CBF occurred in ApoE3 and a ~45% reduction occurred in CBF in ApoE4. This would provide absolute flows of approximately 70 and 55 ml/100g/min in CBF for ApoE3 and ApoE4 mice respectively.

ApoE3 flows are: ~115 ml/100g/min X 0.7 (30% reduction) = 80 ml/100g/min

ApoE4 flows are: ~100 ml/100g/min X 0.55 (45% reduction) = 55 ml/100g/min

This difference in absolute flows is impressive and should be reported. Perhaps a graph of absolute flows over the 4 week period could accompany figure 3A (either text or supplement) or figure S4B.

Response: thank you for this excellent suggestion. We have included a graph illustrating absolute CBF values calculated from the relative reduction by LDF and ASL-MRI baseline CBF data (page 7, lines 160-163; Supplementary Fig. 7B).

Reviewer #2 (Remarks to the Author):

APOE4 is the strongest and most highly replicated genetic risk factor for sporadic Alzheimer's disease (AD). APOE4 increases blood-brain barrier (BBB) damage, cerebral amyloid angiopathy and fibrinogen deposits in the human brain, and has been implicated in increasing oxidative stress in brain by increasing levels of reactive oxygen species (ROS). Furthermore, APOE4 carriers have accelerated pericyte degeneration and BBB breakdown compared to APOE3 carriers, likely dependent upon the cyclophilin A/matrix metalloproteinase-9 pathway. In this manuscript, the authors find that TR-ApoE4 mice have reduced basal neocortical cerebral blood flow (CBF) due to reduced vascular density. CBF change in response to whisker stimulus or application of endothelial-specific vasodilators acetylcholine and bradykinin was also reduced in TR-ApoE4 mice, which could be rescued by application of ROS scavengers to the cortex. Next, the authors induced bilateral common carotid artery stenosis in TR-APOE mice and found increased white matter damage and reduced microvascular flow causing local hypoxia in TR-ApoE4 compared to TR-ApoE3 mice. This is a very interesting paper with cutting-edge methodologies and with potential great impact on better understanding complex cerebrovascular effects of apoE4. Here, we offer several comments that may significantly strengthen this manuscript.

Response: we thank the referee for the supportive comment.

Major comments:

- *As the authors acknowledge in the discussion, the main weakness of this paper is the lack of mechanism of how white matter damage and reduced microvascular density is occurring in TR-ApoE4 mice. Previous studies have shown reduced pericyte coverage in APOE4 human carriers (Lancet Neurol. 2011 Mar;10(3):241-52.; J Cereb Blood Flow Metab. 2016 Jan;36(1):216-27) and TR-ApoE4 mice (Nature. 2012 May 16;485(7399):512-6). Furthermore, the loss of pericytes leads to CBF reductions and white matter dysfunction (Nat Med. 2018 Mar;24(3):326-337). Therefore, the authors are highly encouraged to measure pericyte coverage in the white matter of TR-ApoE mice, and perhaps make the link to what has been known in the literature.*

Response: As suggested by the reviewer, we assessed pericyte coverage by CD13 immunohistochemistry in WT, ApoE3-TR and ApoE4-TR mice with and without BCAS. We confirmed the reduction in pericyte coverage in naïve ApoE4-TR mice (Supplementary Fig. 2), and, in agreement with a previous study², we found that pericyte coverage was not reduced at 28 days after BCAS (Supplementary Fig. 2). Importantly, BCAS did not reduce further pericyte coverage in ApoE4-TR mice (Supplementary Fig. 2). However, considering the involvement of pericytes in WT integrity³, we state in the discussion that the reduced pericyte coverage could

contribute to the increased susceptibility of the CC of ApoE4-TR mice to hypoperfusion (page 5, lines 112-117; page 8, line 197-199; page 11, lines 272-274 and 277; page 12, lines 278-282; page 13, line 303).

• *The authors state on page 9, top paragraph, that, “with 0.18mm microcoils the lesions are restricted to the WM of the CC, sparing the optic nerve and the hippocampus, which is a critical requirement for cognitive testing 32.” However, the authors performed hippocampal-dependent learning tests (Y-maze and novel object recognition) and found deficits. Therefore the authors’ statement should be revised with this in mind. Furthermore, it might be informative to include behavioral tasks more associated with white matter deficits such as radial arm maze (Stroke. 2007 Oct;38(10):2826-32) and complex running wheel (Nat Med. 2018 Mar;24(3):326-337).*

Response: We thank the reviewer for raising this important issue. We agree that the statement “...which is critical for cognitive testing” is misleading in this context. Consequently, we have revised this section of the paper and, based on available evidence, have addressed the role of the hippocampus and its neural connection in the cognitive tasks used in the present study (page 10, lines 230-235; page 20, lines 477-482).

Addressing the role of the hippocampus in the cognitive deficits caused by BCAS requires a more in-depth discussion of the cognitive tests that are currently used in this model and their neural substrates. Several tests have been used to assess the cognitive impact of BCAS, including, for example, the Morris water maze, the Y-maze, Barnes maze, radial arm maze, open field, and novel object recognition (NOR)^{4, 5, 6, 7, 8, 9, 10}. We chose to use the Y-maze and NOR tests based on the following considerations. First, these tests are well documented to produce reliable and quantifiable cognitive deficits in BCAS and are commonly used in this model^{4, 11, 12, 13, 14, 15, 16}. Second, these tests exploit the natural tendency of the mice to explore, more closely reflecting the “activities of daily living” of mice^{17, 18}. Third, they do not require escaping an aversive environment, like the Morris water maze, which stresses the mice and confounds the interpretation of the results by introducing the mice’s ability to cope with stress as a variable^{19, 20, 21}. Fourth, they do not require starving the animals as a motivation stimulus, like the radial maze. Motivation through food restriction is not desirable in experimental situations in which there is weight loss, like in BCAS, because the impact of hunger on the mice may differ across experimental groups, resulting in different motivation to seek the food reward¹⁹. Fifth, these tests evaluate both spatial (Y-maze) and non-spatial working memory (NOR), thereby probing two different cognitive domains. On these bases, we felt that the NOR and Y-maze were well suited for our study.

The reviewer also wonders about the reliance of the Y-maze and NOR on the hippocampus, which is not directly damaged by BCAS, at least if 0.18 mm coils are used^{5, 22}. While it is well established that the hippocampus is critical for cognitive tasks involving memory functions, the integrity of WM connections between the hippocampus and other brain regions are also necessary²³. For example, spatial memory tests like the Morris water maze, radial arm maze, and Y maze are all hippocampal dependent²⁴, but also rely on the continuity of WM tracts connecting the hippocampus with the retrosplenial, prefrontal, and perirhinal cortex, as well as anterior thalamus and other brain regions^{23, 25}. This is thought to be the reason why the Morris water maze, Y-maze and the radial arm maze are altered by BCAS in the absence of overt hippocampal damage^{5, 22}. The specific involvement of the hippocampus in non-spatial working memory tests such as the NOR is unclear, and experiments demonstrating hippocampal involvement have been challenged based on the fact that the lesion methods used also damage the adjacent perirhinal cortex and outflow WM tracts to other cortical regions^{17, 26, 27}. This may explain why the NOR is so effective and widely used in detecting memory deficits in models of cerebral hypoperfusion^{4, 11, 12, 13, 14, 15, 16}.

As for the complex running wheel, this is a sensitive test to assess motor deficits, first applied to a model of corpus callosum agenesis²⁸. However, this test is also altered in models

of Parkinson's and Huntington's disease^{29,30} and, as such, is not WM specific. Since the focus of our study was on cognitive impairment and we did not observe overt motor deficits in our mice (Supplementary Fig. 10), we did not consider necessary using this test.

In summary, the cognitive tasks we used (NOR and Y-maze test): (a) offer the advantage of exploiting the natural behavior of the mice and of not requiring food restriction or an aversive environment, (b) are not different from the radial arm maze test in term of their reliance on hippocampal connectivity - the NOR being the least dependent on the hippocampus, and (c) are exquisitely sensitive to the cognitive impairment induced by BCAS. We have made these points clear in the text (page 9, line 206-207; page 10, lines 230-235; page 20, lines 477-482).

• If possible, the authors should consider adding quantification of white matter hyperintensities using Fazekas scale (Neurology. 1993 Sep;43(9):1683-9).

Response: The Fazekas score provides a semiquantitative assessment of lesion burden in MRI scans of patients³¹. It consists of a 1-3 ordinal scale for periventricular white matter (0 = absence, 1 = "caps" or pencil-thin lining, 2 = smooth "halo," 3 = irregular periventricular hyperintensities extending into the deep white matter) and deep white matter (0=absence, 1 = punctate foci, 2 = beginning confluence of foci, 3 = large confluent areas). This approach is not well suited to accurately quantify small lesions in the mouse corpus callosum. Therefore, we directly assessed white matter lesions in brain sections using image analysis of the Kluver-Barrera white matter stain, the SMI320/myelin basic protein (MBP) ratio (an index of myelination and white matter organization), and the intensity of myelin associated glycoprotein (MAG) (an index of myelination) (Fig. 5). These widely-used independent measures provide a more quantitative assessment of the white matter lesions.

• Did the authors consider measuring changes to oligodendrocytes that may lead to white matter loss? The link pericyte loss, BBB WM breakdown, fibrinogen, oligodendrocytes cell deaths has been established. Perhaps, please consider it for discussion.

Response: Thank you for this suggestion. In new experiments, we investigated the effect of BCAS on oligodendrocytes using Olig2 immunohistochemistry. We found that BCAS reduces Olig2 staining in the CC, an effect more marked in ApoE4-TR than in WT or ApoE3-TR mice, reflecting the increased white matter damage in ApoE4-TR mice (Supplementary Fig. 9). As suggested, we have mentioned in the discussion the potential link between pericyte loss, BBB breakdown, fibrinogen entry and oligodendrocyte damage (page 12, lines 278-282).

• Of relevance to the paper, impaired cerebral blood vessel reactivity to CO2 has been shown in the hippocampus of young, non-demented APOE4 carriers by reduced CBF velocity in the middle cerebral artery using transcranial Doppler (J. Am. Geriatr. Soc. 63, 276–281 2015) and by BOLD-fMRI in response to a memory task (Alzheimers Dement. 11, 648-657.e1 2015). The authors may consider discussing these observations.

Response: Thank you for this suggestion. We have mentioned the hypercapnia reference in the paper (page 9, lines 220-221).

• To provide a complete picture to the readers, the authors perhaps may consider briefly mentioning in the Introduction that APOE isoforms differentially affect the binding and trans-vascular clearance of A β , with impaired clearance in APOE4 carriers. (J Clin Invest. 2008 Dec;118(12):4002-13).

Response: This has been done (page 3, line 71).

• *The three-photon vascular diameter and RBC velocity data are interesting. It would greatly strengthen the manuscript to show that the authors are indeed imaging from the CC, as it is difficult to tell from the 3D volume representation in figure 4A. It would also be helpful to better clarify what depth the authors consider the top of the CC, since in figure 4A the authors mark depths of approx. 870-1050um as CC, but indicate in the figure 4 caption that CC RBC velocity measurements were made between 700-1000um.*

Response: In order to image the CC we relied on: (a) the pattern of the third harmonic generation (THG) signal, which indicated the transition between the looser white matter (WM) tracts of the deep cortical layers and the compact fiber structure of the CC, and (b) the pattern of the FITC dextran signal showing linear microvessels accompanying WM tracts and reduced vascular density. All measurements were made between 800 and 1000 μM , typically $\approx 900 \mu\text{M}$. This variability is due to (a) the inter-animal differences in the distance between the pial surface and the CC and (b) the cortical thickness at the imaging site within the cranial window which, varies side-to-side and rostro-caudally. We have made these points clear in the text (page 17, lines 424-429).

• *For the two-photon measurements of RBC velocity (presented in Fig. 1), it would be helpful to clarify over what range of depths the measurements were made, and if the range of depths were similar for all the animals studied.*

Response: Two-photon measurements of RBC velocity (Fig. 1B) were made at the depth of 200 μm in all groups of mice. We have added this information to the methods (page 17, line 419).

Minor comments

• *For bar graphs with multiple comparisons (more than 2 bars), it would be helpful for the reader if the authors add lines/indicators to distinguish which group comparisons are significant. See for example Figure 1A, is ApoE4 significant compared to either WT, ApoE3 or both? Adding this visual cue would be helpful.*

Response: Thank you for this suggestion to increase the clarity of the statistics in the figures. However, in figure 1A,C adding lines clutters the figure and we elected to use symbols instead, the asterisk indicating significance from both the ApoE3 and WT group (as now clearly indicated in the figure legend). We have also specified that there is no statistical difference between the WT and the ApoE3 group. For consistency, we have used symbols in all figures to indicate statistical differences. We hope that this solution satisfies the reviewer and the editors.

• *The authors should consider including a table of all antibodies used in the supplementary materials.*

Response: This has been done (Supplementary Table 1).

• *It would also be helpful to provide more details of the imaging parameters, methods, and depth ranges used for two- and three-photon imaging in the methods. At the least, it would be helpful to provide an explanation of what the "custom written MATLAB code" for measurement of RBC velocity does. Is this a variant of one of the published velocity measurement softwares? Similarly, it would be beneficial to provide more details of the vessel diameter measurement methodology.*

Response: We have used well established methods from our laboratories that have been published^{32, 33}. We have now referred to these papers (page 18, line 431). In addition, we have expanded the method section related to imaging and indicated the measuring depths (page 17, lines 420-427; page 18, lines 428-431).

• *In figure 1B, it would be helpful to indicate the depth over which the vascular projection images*

were made. Also, in the plot of velocity vs. diameter, it would be helpful to provide some kind of lines or marks to better indicate the capillary diameter range. Since the capillary diameter range is <10µm and venule range is >11 µm, are the venule-side vessels between 10-11µm diameter shown on the plot?

Response: We have provided the depth of vascular projections images (200µm) and have indicated that capillaries were considered to be ≤10µm in diameter. Vertical lines were added to Fig. 1B to set the boundaries between capillaries and arterioles/venules (Figure 1 legend).

• Please provide an example of a THG channel image used for RBC velocity measurements separate from the dextran dye. The merged images in Fig. 4 are colorful, but make it more difficult to distinguish the THG signal.

Response: The attribution of the source of the THG signal to RBC is well established and has been defined in previous publications^{34, 35}. However, we agree with the reviewer that it would be desirable to provide more detail since the readership of the journal may not be familiar with this technology. To this end we added a new figure showing the relationships between the THG and dextran signals (Supplementary Fig. 11).

• In figure 3B, the color scale bar does not match the color scale used in the images.

Response: Thank you, this oversight was corrected.

• Figure 5B would be more convincing with better quality representative images.

Response: This has been done.

• In figure 4G, the data does not look normally distributed. Was it tested for normality before running statistics?

Response: The reviewer is correct, and we have used a the Kruskal-Wallis test, a non-parametric test which does not require a normally distributed data set, to test for statistical significance. This has now been indicated in the figure legend (Fig 4G) and in the methods (page 21, lines 507-508 and 510-511; page 31, line 841).

• Line 223: "vascular density" may be a more descriptive term than "vascularity."

Response: we have used vascularity instead of vascular density (page 10, lines 250)

• The manuscript should be checked carefully for typos, e.g. "SMI32," "ApoE4=TR," "box plot," incorrect supplemental figure reference in Methods.

Response: that you for pointing out these oversights which have been corrected (page 8, line 193; Fig. 3 legend; Fig. 4 legend).

Reviewer #3 (Remarks to the Author):

While apoE4 is an important risk factor for both Alzheimers disease and vascular dementia most of the apoE4 in vivo mice model studies to date focus on the neuronal and inflammatory effects of apoE4. This missing link is provided by the present study which focuses on the vascular effects of apoE4 in targeted replacement mice. The results obtained first revealed that under resting conditions the cerebral blood flow of the apoE4 mice is reduced relative to that of the apoE3 mice and that this is associated with a reduction in the number and area of the microvessels. The study than focused onthe neurovascular unit and the coupling between

neuronal excitation and cerebral blood flow. This revealed that, like under basal conditions, the increase in cerebral blood flow in the somatosensory cortex following whisker stimulation was markedly reduced in the apoE4 mice. Furthermore, stenosis induces cerebral hypoperfusion in apoE4 mice which is associated with more severe white matter hypoxia, myelin disruption and cognitive impairments. The present experimental design in which the authors look at the vasculature at several complimentary levels including basal and stimulated conditions as well as under stress challenges is comprehensive and impressive and adds significantly to the impact of the observations. This is further fortified by the consistency of the observations regarding the vascular effects of apoE4 under the different conditions. The interplay between the neuronal and vascular effects of apoE4 is still an open question regarding which the presently studied experimental system could shed some light.

Response: Thank you for the supportive comments.

For example, it would be of interest to assess the extent to which the whisker barrel stimulation vascular impairments in the apoE4 mice are associated with corresponding cortical neuronal impairments. This is of particular interest as the response of the vasculature to pharmacological treatments (eg the NO donor SNAP or the smooth muscle relaxant adenosine) are not affected by apoE4.

Response: The referee raises the important point of whether the attenuation of the CBF response to whisker stimulation in ApoE4-TR mice is related to decreased neuronal activation. To address this question, in new experiments we used Ca²⁺ imaging to investigate the changes in intracellular Ca²⁺ induced glutamate receptor activation, a major driver of neural activity in the whisker barrel cortex³⁶. We found that the glutamatergic agonist NMDA (40 μM) elicited comparable increases in intracellular Ca²⁺ in isolated neocortical neurons of WT, ApoE3-TR and ApoE4-TR mice (Supplementary Fig. 4). Furthermore, as suggested, we examined the immunoreactivity of the pre-synaptic marker synaptophysin and no differences were observed between WT, ApoE3 and ApoE4-TR mice (Supplementary Fig. 5). These observations, collectively, indicate that the attenuation of functional hyperemia in ApoE4-TR mice is unlikely to result from a reduction in the ability of neurons to respond to glutamatergic activation or pre-synaptic input. These new findings are presented in the text (page 6, lines 137-144).

In addition, the authors should include immunohistochemical data of the effects of apoE4 on neuronal parameters such as synaptophysin and if possible on microglial parameters such as Iba1. Since previous studies have focused primarily on the neuronal and inflammatory effects of apoE4 we feel that the impact and significance of this study will be markedly increased by such additions.

Response: We have also performed Iba1 immunohistochemistry for microglia/macrophages in the CC and found no differences in the intensity of the stain in naive WT, ApoE3-TR and ApoE4-TR mice. As previously reported³⁷, BCAS increased Iba1 immunostain in WT, ApoE3-TR and ApoE4-TR mice. However, consistent with the increased WM damage, the Iba1 signal was enhanced in ApoE4-TR compared to WT and ApoE3-TR mice, suggesting increased microglial activation. These new findings have been reported in new supplementary figures (Supplementary Fig. 8) and described in the text (page 8, lines 193-195).

Please add the age and gender of the mice. These technical points are important in order to establish how early the observed vascular pathology evolves and in view of the fact that in man and mice females are more susceptible than males to apoE4.

Response: The age and sex of the mice is indicated (page 14, lines 329-330).

References

1. Shen Q, Duong TQ. Magnetic Resonance Imaging of Cerebral Blood Flow in Animal Stroke Models. *Brain Circ* **2**, 20-27 (2016).
2. Liu Q, *et al.* Experimental chronic cerebral hypoperfusion results in decreased pericyte coverage and increased blood-brain barrier permeability in the corpus callosum. *J Cereb Blood Flow Metab*, 271678X17743670 (2017).
3. Montagne A, *et al.* Pericyte degeneration causes white matter dysfunction in the mouse central nervous system. *Nat Med* **24**, 326-337 (2018).
4. Patel A, *et al.* Chronic cerebral hypoperfusion induced by bilateral carotid artery stenosis causes selective recognition impairment in adult mice. *Neurol Res* **39**, 910-917 (2017).
5. Shibata M, *et al.* Selective impairment of working memory in a mouse model of chronic cerebral hypoperfusion. *Stroke* **38**, 2826-2832 (2007).
6. Nishio K, *et al.* A mouse model characterizing features of vascular dementia with hippocampal atrophy. *Stroke* **41**, 1278-1284 (2010).
7. Hase Y, *et al.* The effects of environmental enrichment on white matter pathology in a mouse model of chronic cerebral hypoperfusion. *J Cereb Blood Flow Metab* **38**, 151-165 (2018).
8. Higaki A, *et al.* Beneficial Effect of Mas Receptor Deficiency on Vascular Cognitive Impairment in the Presence of Angiotensin II Type 2 Receptor. *Journal of the American Heart Association* **7**, (2018).
9. Saggi R, *et al.* Astroglial NF- κ B contributes to white matter damage and cognitive impairment in a mouse model of vascular dementia. *Acta Neuropathol Commun* **4**, 76 (2016).
10. Wolf G, *et al.* Differentially Severe Cognitive Effects of Compromised Cerebral Blood Flow in Aged Mice: Association with Myelin Degradation and Microglia Activation. *Front Aging Neurosci* **9**, 191 (2017).
11. Dominguez R, *et al.* Estradiol Protects White Matter of Male C57BL6J Mice against Experimental Chronic Cerebral Hypoperfusion. *J Stroke Cerebrovasc Dis* **27**, 1743-1751 (2018).
12. Khan MB, *et al.* Remote ischemic postconditioning: harnessing endogenous protection in a murine model of vascular cognitive impairment. *Transl Stroke Res* **6**, 69-77 (2015).
13. Lee ES, Yoon JH, Choi J, Andika FR, Lee T, Jeong Y. A mouse model of subcortical vascular dementia reflecting degeneration of cerebral white matter and microcirculation. *J Cereb Blood Flow Metab*, 271678X17736963 (2017).

14. Mehla J, Lacoursiere S, Stuart E, McDonald RJ, Mohajerani MH. Gradual Cerebral Hypoperfusion Impairs Fear Conditioning and Object Recognition Learning and Memory in Mice: Potential Roles of Neurodegeneration and Cholinergic Dysfunction. *J Alzheimers Dis* **61**, 283-293 (2018).
15. Toyama K, *et al.* Apoptosis signal-regulating kinase 1 is a novel target molecule for cognitive impairment induced by chronic cerebral hypoperfusion. *Arterioscler Thromb Vasc Biol* **34**, 616-625 (2014).
16. Zuloaga KL, *et al.* High fat diet-induced diabetes in mice exacerbates cognitive deficit due to chronic hypoperfusion. *J Cereb Blood Flow Metab* **36**, 1257-1270 (2016).
17. Bevins RA, Besheer J. Object recognition in rats and mice: a one-trial non-matching-to-sample learning task to study 'recognition memory'. *Nat Protoc* **1**, 1306-1311 (2006).
18. Dellu F, Contarino A, Simon H, Koob GF, Gold LH. Genetic differences in response to novelty and spatial memory using a two-trial recognition task in mice. *Neurobiol Learn Mem* **73**, 31-48 (2000).
19. Vorhees CV, Williams MT. Assessing spatial learning and memory in rodents. *ILAR journal / National Research Council, Institute of Laboratory Animal Resources* **55**, 310-332 (2014).
20. Wenk GL. Assessment of spatial memory using the radial arm maze and Morris water maze. *Curr Protoc Neurosci* **Chapter 8**, Unit 8 5A (2004).
21. Puzzo D, Lee L, Palmeri A, Calabrese G, Arancio O. Behavioral assays with mouse models of Alzheimer's disease: practical considerations and guidelines. *Biochem Pharmacol* **88**, 450-467 (2014).
22. Coltman R, *et al.* Selective white matter pathology induces a specific impairment in spatial working memory. *Neurobiol Aging* **32**, 2324 e2327-2312 (2011).
23. Aggleton JP, Pearce JM. Neural systems underlying episodic memory: insights from animal research. *Philos Trans R Soc Lond B Biol Sci* **356**, 1467-1482 (2001).
24. Becker JT, Walker JA, Olton DS. Neuroanatomical bases of spatial memory. *Brain Res* **200**, 307-320 (1980).
25. Whishaw IQ, Jarrard LE. Similarities vs. differences in place learning and circadian activity in rats after fimbria-fornix section or ibotenate removal of hippocampal cells. *Hippocampus* **5**, 595-604 (1995).
26. Cohen SJ, Stackman RW, Jr. Assessing rodent hippocampal involvement in the novel object recognition task. A review. *Behav Brain Res* **285**, 105-117 (2015).
27. Mumby DG. Perspectives on object-recognition memory following hippocampal damage: lessons from studies in rats. *Behav Brain Res* **127**, 159-181 (2001).

28. Schalomon PM, Wahlsten D. Wheel running behavior is impaired by both surgical section and genetic absence of the mouse corpus callosum. *Brain Res Bull* **57**, 27-33 (2002).
29. Liebetanz D, Baier PC, Paulus W, Meuer K, Bahr M, Weishaupt JH. A highly sensitive automated complex running wheel test to detect latent motor deficits in the mouse MPTP model of Parkinson's disease. *Exp Neurol* **205**, 207-213 (2007).
30. Mandillo S, *et al.* Early motor deficits in mouse disease models are reliably uncovered using an automated home-cage wheel-running system: a cross-laboratory validation. *Dis Model Mech* **7**, 397-407 (2014).
31. Fazekas F, Chawluk JB, Alavi A, Hurtig HI, Zimmerman RA. MR signal abnormalities at 1.5 T in Alzheimer's dementia and normal aging. *AJR Am J Roentgenol* **149**, 351-356 (1987).
32. Kim TN, *et al.* Line-scanning particle image velocimetry: an optical approach for quantifying a wide range of blood flow speeds in live animals. *PLoS One* **7**, e38590 (2012).
33. Santisakultarm TP, *et al.* In vivo two-photon excited fluorescence microscopy reveals cardiac- and respiration-dependent pulsatile blood flow in cortical blood vessels in mice. *Am J Physiol Heart Circ Physiol* **302**, H1367-1377 (2012).
34. Dietzel S, *et al.* Label-free determination of hemodynamic parameters in the microcirculation with third harmonic generation microscopy. *PLoS One* **9**, e99615 (2014).
35. Farrar MJ, Wise FW, Fetcho JR, Schaffer CB. In vivo imaging of myelin in the vertebrate central nervous system using third harmonic generation microscopy. *Biophys J* **100**, 1362-1371 (2011).
36. Feldmeyer D, *et al.* Barrel cortex function. *Prog Neurobiol* **103**, 3-27 (2013).
37. Hattori Y, *et al.* A novel mouse model of subcortical infarcts with dementia. *J Neurosci* **35**, 3915-3928 (2015).

REVIEWERS' COMMENTS:

Reviewer #1 (Remarks to the Author):

No further comments.

Reviewer #2 (Remarks to the Author):

The authors have addressed all comments.

Reviewer #3 (Remarks to the Author):

The authors have addressed all of this reviewer's comments very eloquently and I recommend that the paper be accepted for publication in its present form.

We wish to thank the reviewers for their careful reading of the manuscript and recommendation for publication.

Reviewer #1 (Remarks to the Author):

No further comments.

Response: We thank the reviewer.

Reviewer #2 (Remarks to the Author):

The authors have addressed all comments.

Response: We thank the reviewer.

Reviewer #3 (Remarks to the Author):

The authors have addressed all of this reviewer's comments very eloquently and I recommend that the paper be accepted for publication in its present form

Response: We thank the reviewer.